# Identification and characterization of a new isoform of small GTPase RhoE

Yuan Dai[1], Weijia Luo[1], Xiaojing Yue[2], Wencai Ma[3], Jing Wang[3] & Jiang Chang[1✉]

The Rho family of GTPases consists of 20 members including RhoE. Here, we discover the existence of a short isoform of RhoE designated as RhoEα, the first Rho GTPase isoform generated from alternative translation. Translation of this new isoform is initiated from an alternative start site downstream of and in-frame with the coding region of the canonical RhoE. RhoEα exhibits a similar subcellular distribution while its protein stability is higher than RhoE. RhoEα contains binding capability to RhoE effectors ROCK1, p190RhoGAP and Syx. The distinct transcriptomes of cells with the expression of RhoE and RhoEα, respectively, are demonstrated. The data propose distinctive and overlapping biological functions of RhoEα compared to RhoE. In conclusion, this study reveals a new Rho GTPase isoform generated from alternative translation. The discovery provides a new scope of understanding the versatile functions of small GTPases and underlines the complexity and diverse roles of small GTPases.

[1] Center for Genomic and Precision Medicine, Texas A&M University College of Medicine, Institute of Biosciences and Technology, Houston, TX 77030, USA. [2] Department of Obstetrics and Gynecology, Nanfang Hospital, Southern Medical University, Guangzhou, China. [3] Department of Bioinformatics and Computational Biology, The University of Texas MD Anderson Cancer Center, Houston, TX 77030, USA. ✉email: jiangchang@tamu.edu

The Rho proteins belong to the Ras superfamily and these small GTPases function as G signaling proteins that convert and amplify external signals into cellular effects[1,2]. Rho GTPases have shown to regulate many aspects of intracellular actin dynamics, such as cytoskeleton rearrangement, cell migration, vesicular trafficking, cell polarity, cell cycle, and transcriptional dynamics[3–5]. In mammals, the Rho family contains 20 members[1]. Though the majority of them are respectively activated and deactivated by the GTP-bound and GDP-bound cycling, "atypical" GTPases in the Rnd subgroup are unusual as they are constitutively GTP bound[6,7]. Therefore, it is not regulated by GTP/GDP cycling, but by the expression level and by protein modifications, such as prenylation and phosphorylation[4]. There are three members, Rnd1, Rnd2, and Rnd3/RhoE, in the Rnd subgroup and Rnd3/RhoE is the most extensively studied member. The current paradigm, derived from the majority of the published studies on RhoE, is that RhoE functions as a repressor that directly inhibits the RhoA effector Rho-associated coiled-coil kinase 1 (ROCK1)[8,9]. While the regulatory roles of RhoE have been almost entirely linked to cell actin cytoskeleton dynamics, cell migration, and apoptosis in its early studies[9–13], we have uncovered versatile functions of RhoE in intracellular $Ca^{2+}$ homeostasis regulation[14], apoptosis[15], angiogenesis[16], inflammation[17], brain development[18], and glioblastoma genesis[19,20] along with other groups[21–24]. All of these studies suggest a broad spectrum of biological and pathological functions of RhoE.

In this study, we discovered the existence of a new short isoform of RhoE, which we designated as RhoEα. Using multiple approaches, we identified a new alternative translation start site responsible for the expression of RhoEα protein, and this site is located downstream of the classical RhoE translation start site. The RhoEα protein characterizations, including subcellular distribution and protein stability were defined. Interactions of RhoEα to RhoE effectors were determined as well. Finally, the potential differential functions of the two RhoE isoforms were investigated and compared using RNA-seq approaches. While new isoforms derived from alternative splicing were identified in Rho GTPase family before[25,26], we report here that RhoEα is the first Rho GTPase isoform generated from alternative translation, demonstrating the diversity of the GTPase family. The study opens a new perspective for future studies in the exploration of Rho GTPase multiplicity and the interpretation of their diverse functions.

## Results

### Two protein bands were detected by anti-RhoE antibody.
During the studies of biological functions of atypical small GTPase RhoE, we often detected two immunoblot bands for RhoE in animal tissues and various cell lines[17]. Figure 1a showed a representative immunoblot of RhoE protein expression in mouse heart, brain, lung, and liver. The two immunoblot protein bands were detected in HeLa, HEK293, and mouse embryonic fibroblast (MEF) cells as well (Fig. 1b), and their expression levels can be decreased by the treatment of siRNA specific for RhoE in these cells (Fig. 1b). We next generated three HeLa cell lines of RhoE knockout by CRISPR, and observed a complete elimination of the two immunoblot bands in all three cell lines (Fig. 1c). To validate the knockout result in vivo, we assessed RhoE expression in RhoE null MEF cells isolated from RhoE global knockout mice[14], and found that genetic silencing of RhoE completely removed the two immunoblot bands (Fig. 1d), suggesting that both protein bands are highly associated with RhoE. We hypothesize and want to demonstrate the existence of a new RhoE isoform.

To determine if the expression of two RhoE immunoblot bands shared the same transcription promoter, we introduced Cas9-

synergistic activation mediator (Cas9-SAM) along with the sgRNA specifically targeting RhoE promoter. The result showed a dramatic increase in both immunoblot bands when RhoE promoter was transcriptionally activated (Fig. 1e), indicating that the two immunoblots of RhoE were transcriptionally regulated by the same promoter and were both originated from the RhoE gene.

To provide definitive evidence, mass spectrometry analysis was performed in two protein bands ~27 kDa immunoprecipitated by RhoE antibody (Fig. 1f). The mass sequences of the two protein bands highly matched to RhoE protein, compatible with the existence of two isoforms of RhoE.

### The short isoform of RhoE was translated from ATG[46].
Alternative translation initiation and alternative splicing from a single gene are the two major mechanisms resulting in a generation of protein isoform in most circumstances[27–30]. To determine whether these mechanisms are responsible for existence of the additional RhoE protein band, we first generated RhoE knockout cell line by CRISPR technique in HeLa cells, and then an expression vector containing human RhoE 5′ untranslated region (UTR) and coding sequence was transiently transfected into this cell line. We detected the expression of both RhoE and the additional band in the transient transfection cells (Fig. 2a). Molecular weights of the two proteins were consistent with endogenous RhoE immunoblot bands (Fig. 2a). The result was further confirmed by a second expression vector containing the same human RhoE cDNA sequence fused with a flag tag. Two protein bands of RhoE were shown again by immunoblot using anti-flag antibody (Fig. 2b). Since the expression of two RhoE proteins directly came from the expression vector, the result ruled out the possibility of alternative splicing mechanism responsible for this undefined protein.

To examine whether alternative translation initiation generates this additional RhoE-like protein, we looked for possible alternative translation initiation site (aTIS) in RhoE gene. We first used a strategy by inserting a translation stop codon immediate before and after the currently known RhoE TIS called ATG[1], respectively, in human RhoE expression vector (Fig. 3a, left panel). The expression of RhoE by these two expression vectors was analyzed by western blot. We found that insertion of stop codon before ATG[1] showed no effect on the expression of both RhoE proteins (Fig. 3a, right panel lane 3), indicating that 5′ UTR of RhoE did not harbor any alternative translation start site. Although insertion of the stop codon after ATG[1] eliminated the upper protein band (Fig. 3a, right panel lane 4), which is the known RhoE protein, the lower band still existed. The result clearly indicates the existence of a new RhoE isoform and translation of this isoform starts downstream of ATG[1]. To distinguish these two proteins, we designated the new isoform of RhoE as RhoEα.

We then reviewed human RhoE mRNA coding region and found five additional ATG codons in-frame with ATG[1]. They were ATG[46], ATG[388], ATG[493], ATG[658], and ATG[730] (Fig. 3b). Among them, only ATG[46] was possible to encode a protein with a molecular weight close to the detected band. We tested this assumption by mutating ATG[1] and ATG[46], respectively, in RhoE expression vector (Fig. 3c). Again, mutation of ATG[1] to AAA resulted in a loss of the RhoE protein expression (Fig. 3c, right panel lane3). The replacement of ATG[46] with AAA, however, eliminated RhoEα protein expression (Fig. 3c, right panel lane 4), further confirming our prediction.

To demonstrate the result in ex vivo, we individually deleted ATG[1] and ATG[46] in RhoE genome via CRISPR-mediated point mutation technique in HeLa cells (Fig. 3d). Again, the result was consistent with the above in vitro observation, strongly

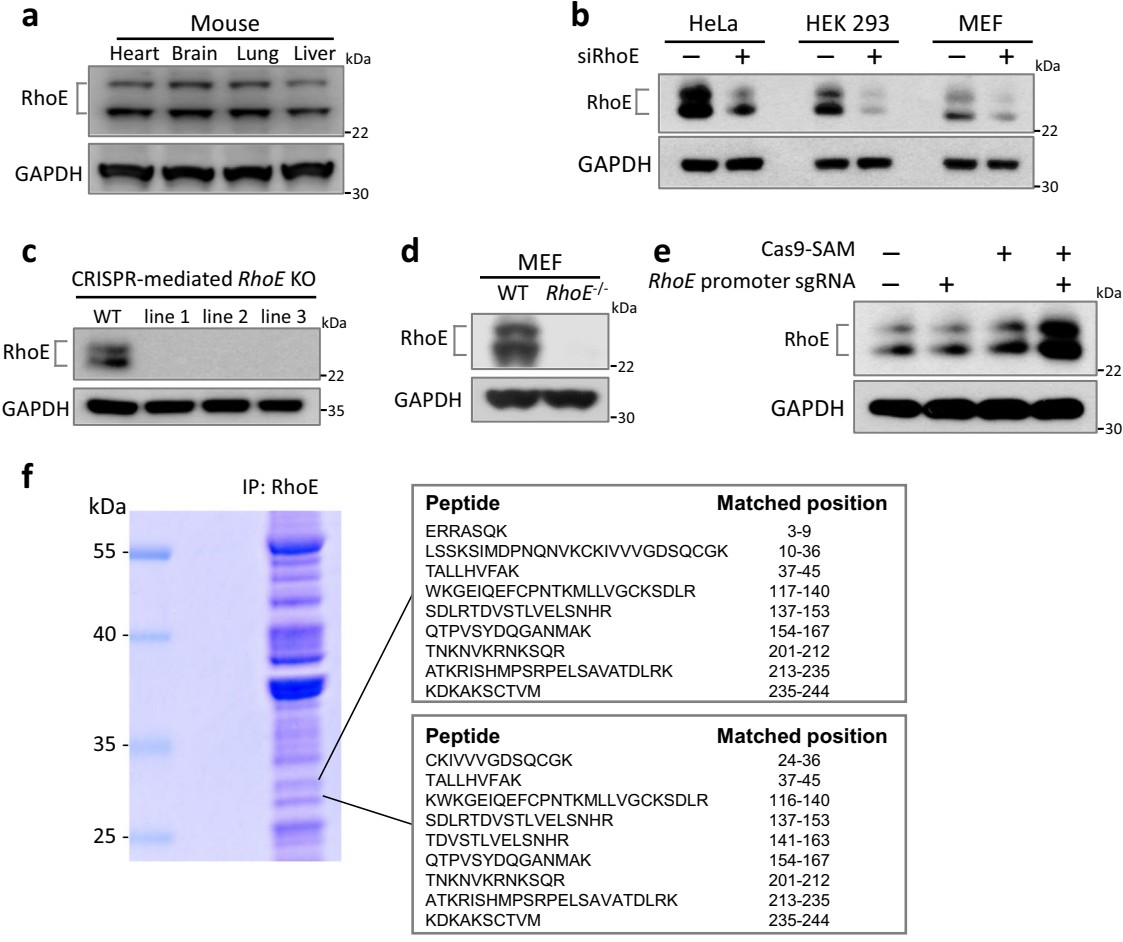

**Fig. 1 Two specific proteins are detected by RhoE antibody. a** Immunoblot for RhoE in wild-type mouse tissues. **b** Expression of RhoE in normal and RhoE knockdown cell lines. Indicated cells were transfected with either nontargeting siRNA or RhoE-specific siRNA. **c** Immunoblot for RhoE in wild-type and CRISPR-mediated *RhoE* knockout (*RhoE* KO) HeLa cells. Three *RhoE* KO cell clones were presented. **d** Immunoblot for RhoE in MEF cells from wild-type and *RhoE* null (*RhoE*$^{-/-}$) mice. **e** Transcriptional activation of *RhoE* in HeLa cells by Cas9-synergistic activation mediator (SAM). *RhoE* promoter-specific sgRNA was transfected along with the Cas9-SAM components in HeLa cells. Cell lysates were immunoblotted for RhoE. **f** Protein pull-down using RhoE antibody in HEK 293 cells, followed by LC–MS/MS assay. Peptides identified from the upper and lower gel band were aligned to RhoE protein sequence.

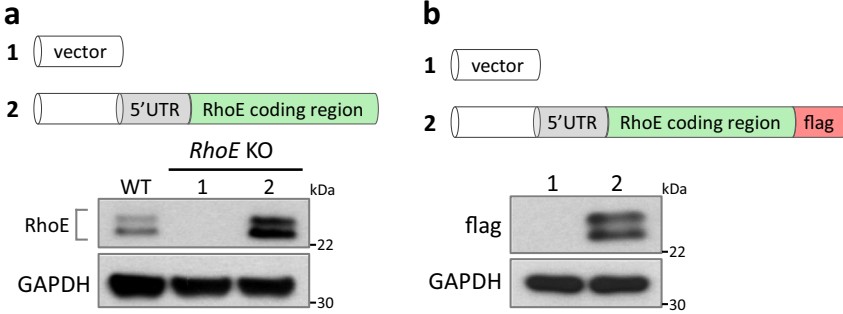

**Fig. 2 A new translation initiation site is responsible for the expression of RhoE isoform. a** Immunoblot for RhoE in wild-type and *RhoE* knockout (*RhoE* KO) HeLa cells transfected with either the empty vector or the RhoE expression construct. **b** 5′ UTR and coding region of RhoE was constructed into the C-flag vector. HeLa cells were transfected with the indicated plasmids. Cell lysates were immunoblotted for flag-tagged proteins.

supporting ATG[1] and ATG[46] as the TIS responsible for RhoE and RhoEα expression, respectively.

We noticed that a computer-predicted aTIS, CTG[−81], at the 5′ UTR of *RhoE* was reported[31]. We examined this aTIS and found that the expression of RhoEα was not interfered after deletion of this presumed aTIS (Supplementary Fig. 1).

Collectively, these data uncover RhoEα as a 15 amino acids shorter isoform of RhoE. A second TIS ATG[46] in *RhoE* coding region, but not alternative splicing, is responsible for RhoEα expression.

It is intriguing to realize that a potent Kozak sequence is around ATG[46], while the ATG[1] has less optimal Kozak sequence (Fig. 3e). Our result supports Kozak's observation, which an optimal context around the second AUG could trigger 40 S ribosomal subunits to initiate translation[32]. Finally, a highly conserved ATG[46] among vertebrate species is observed,

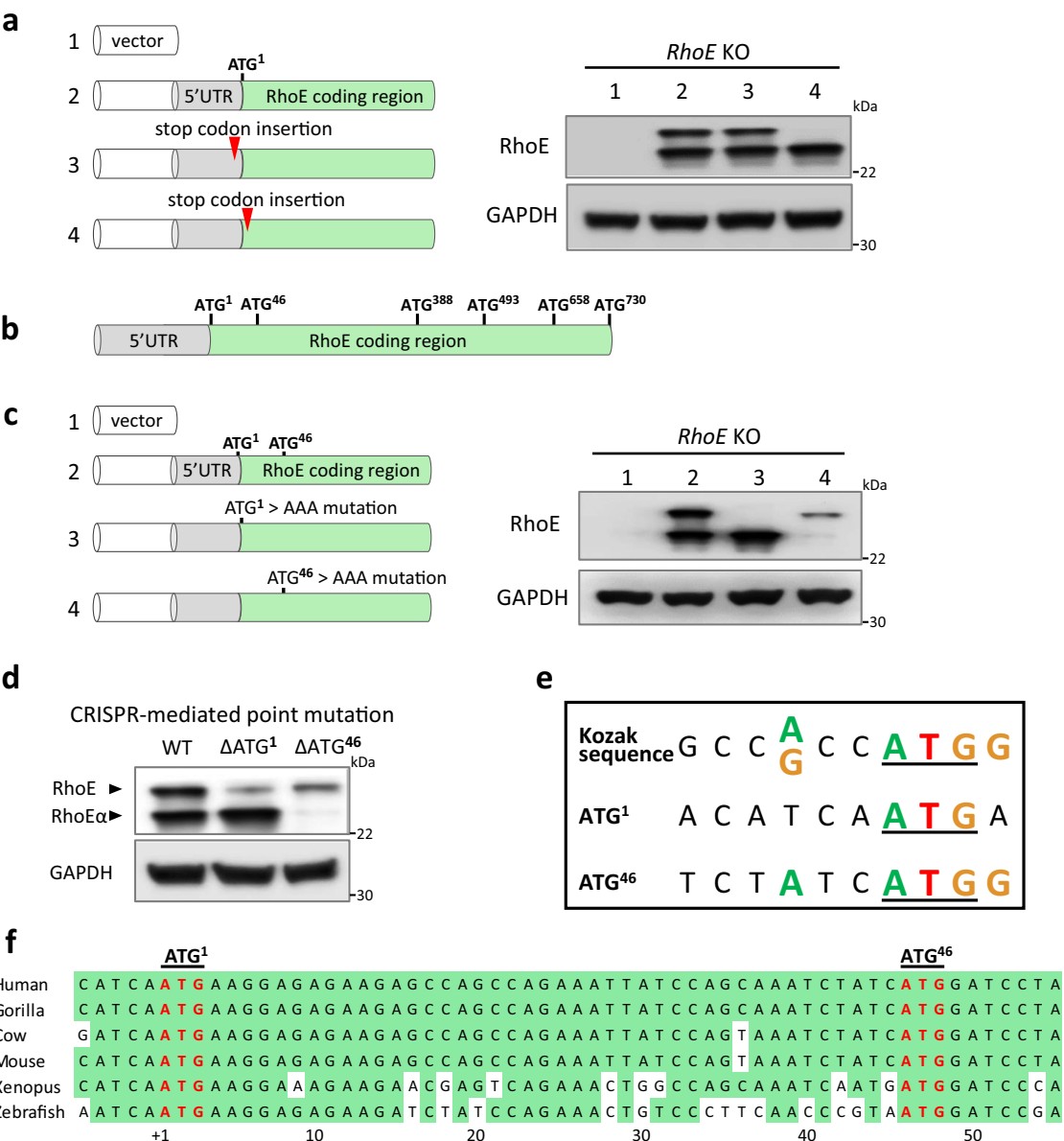

**Fig. 3 Determination of the translation start site for the RhoE isoform. a** RhoE expression vector was mutated by the stop codon insertion before and after ATG[1]. RhoE knockout HeLa cells were transfected with indicated plasmids and cell lysates were immunoblotted. **b** Schematic of potential alternative translation start sites in RhoE coding region. **c** ATG[1] and ATG[46] were individually mutated to AAA. RhoE knockout HeLa cells were transfected with indicated plasmids and cell lysates were immunoblotted. **d** Single-stranded donor oligonucleotides (ssODN) were designed for inducing either ATG[1] or ATG[46] genomic deletion. HeLa cells were co-transfected with the Cas9; RhoE sgRNA plasmid and the indicated ssODN. Cell lysates were immunoblotted. **e** Comparison of Kozak sequences around human RhoE translation initiation sites. The strongest Kozak context promoting translation initiation contains a purine, preferably adenine A, 3 bp before ATG, and a guanine G following ATG. **f** Conservation analysis for RhoEα translation start site among species.

suggesting a broad existence of RhoEα expression across species (Fig. 3f). The result also rules out a possibility of mobility shift caused by a posttranslational modification (PTM)[33].

**Localization and protein stability between RhoE and RhoEα.** We next investigated the characteristics of RhoEα protein. Protein localization is important for protein functions and activities, and RhoE has been reported to localize in the cytosol, plasma membrane, and internal membranes[6,7]. We compared the two proteins' cellular localization by co-transfection of GFP-RhoE and mCherry-RhoEα expression vectors. GFP-RhoE expression vector was validated by western blot to ensure the expression of RhoE only, but not RhoEα (Fig. 4a). Fluorescent images showed that RhoEα shared similar subcellular distribution with RhoE

(Fig. 4b), suggesting that missing the first 15 amino acids of N-terminus of RhoE has a minimal effect on the subcellular localization of RhoEα in a normal situation. To further confirm this observation, we also compared the cellular localization of GFP-RhoE ATG[46] > AAA mutant to the localization of RhoEα protein expressed by mCherry-RhoEα. Again, the result is consistent and supports the conclusion (Supplementary Fig. 2).

We often observed a higher expression level of RhoEα compared to RhoE in tissues and cells (Fig. 1), which intrigued us to evaluate the difference in their protein stability. We conducted cycloheximide (CHX) chase analysis and found that RhoEα exhibited significantly longer half-life than RhoE (Fig. 4c), consistent with the observed high expression level of RhoEα. The different protein stability between two isoforms indicates

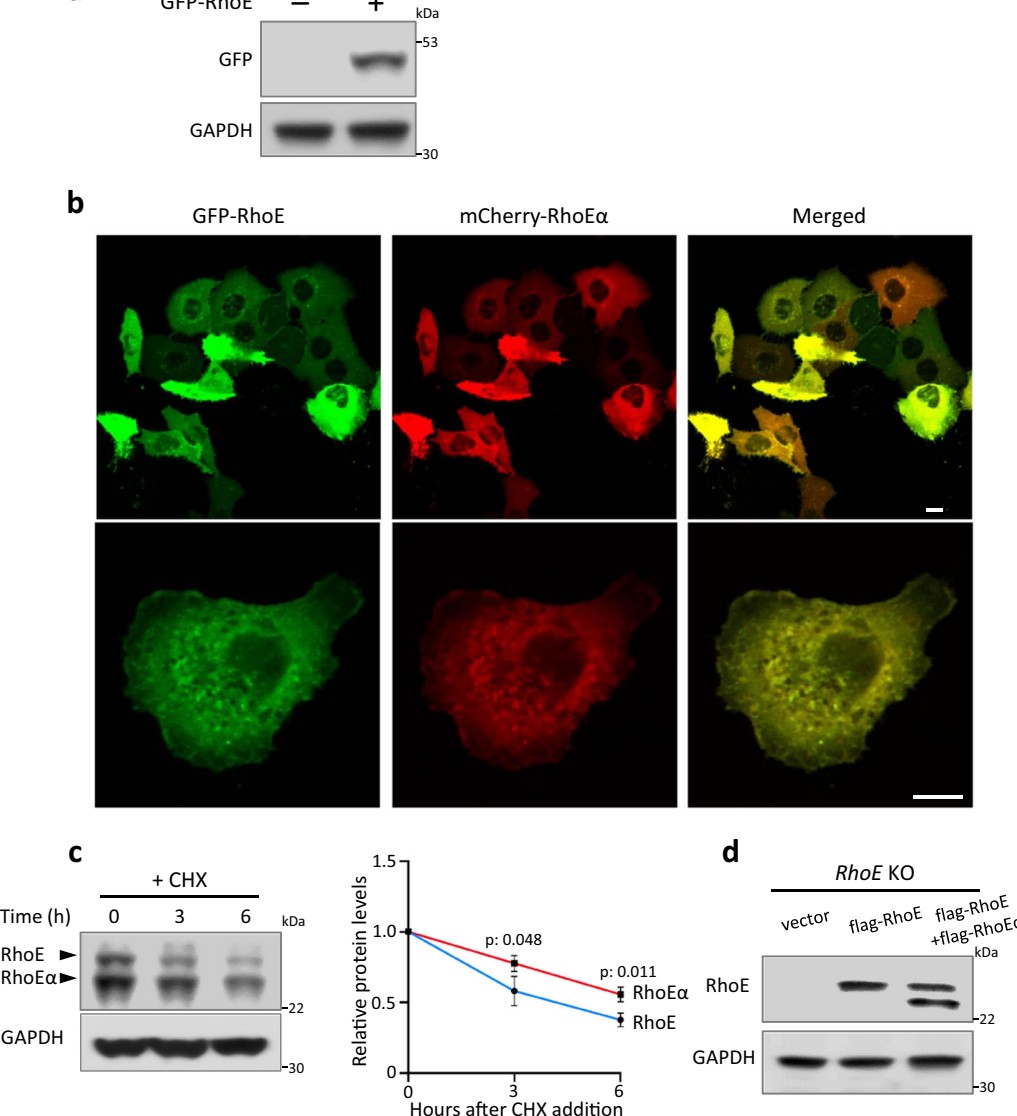

**Fig. 4 Characterizations of RhoEα. a** Immunoblot showed expression of only RhoE, but not RhoEα from the GFP-RhoE expression plasmid. **b** Confocal images exhibited subcellular localizations of GFP-RhoE and mCherry-RhoEα in the transfected HeLa cells. Scale bar: 10 μm. **c** Immunoblot analysis showed the degradation of RhoE and RhoEα along with time in HeLa cells. Cycloheximide (CHX) 20 μg/ml was used to inhibit protein synthesis. Protein levels of RhoE and RhoEα were normalized to GAPDH, and three independent experiments were pooled and quantified (right panel). **d** Indicated plasmids were transfected into *RhoE* knockout (*RhoE* KO) HEK293 cells. Cell lysates were immunoblotted for RhoE.

potential distinct mechanisms in the two protein degradations. To rule out if there is any direct effort of RhoEα on the expression of RhoE or RhoE protein stability, we reintroduced RhoEα in the cells and assessed the level of RhoE protein by western blot. We found that forced expression of RhoEα had a minimal impact on RhoE protein level (Fig. 4d).

**Functional similarity and divergence between RhoE and RhoEα.** RhoE was originally found to associate with RhoA/ROCK1 signaling as an antagonist, and led to an altered cell morphology to "round" shape due to the inhibition of actin cytoskeletal assembly[4,7]. We first investigated if RhoEα plays a similar regulatory role in RhoA/ROCK1 signaling through interacting with ROCK1. A co-IP assay was performed, and showed that ROCK1 bound to both RhoE and RhoEα (Fig. 5a). The binding affinity of ROCK1 to the two isoforms was comparable. We next examined the effect of RhoEα on cell actin

assembly. Disrupted cytoskeleton actin filaments were obviously exhibited in both RhoEα and RhoE-transfected cells, resulting in a much small cell size (Fig. 5b).

RhoE often exerts its functions through interacting with other effectors as well. We tested the interaction of the isoforms with two additional well-known RhoE effectors, p190RhoGAP and synectin-binding RhoA exchange factor (Syx), which were involved in the regulation of cell protrusion and migration[34], and embryonic cell shape[35], respectively. Again, the co-IP assays indicated that RhoE and RhoEα bound to the two effectors (Fig. 5c), suggesting a possible functional redundancy of the two isoforms.

Considering recently unrevealed diverse functions of RhoE[4,36], we further compared gene expression profiles and the associated pathways in two proteins, using unbiased approaches. We first established two cell lines with a *RhoE* null background in HEK 293 and HeLa cells, respectively, via CRISPR-mediated *RhoE* gene knockout. The two cell lines were chosen due to their wide

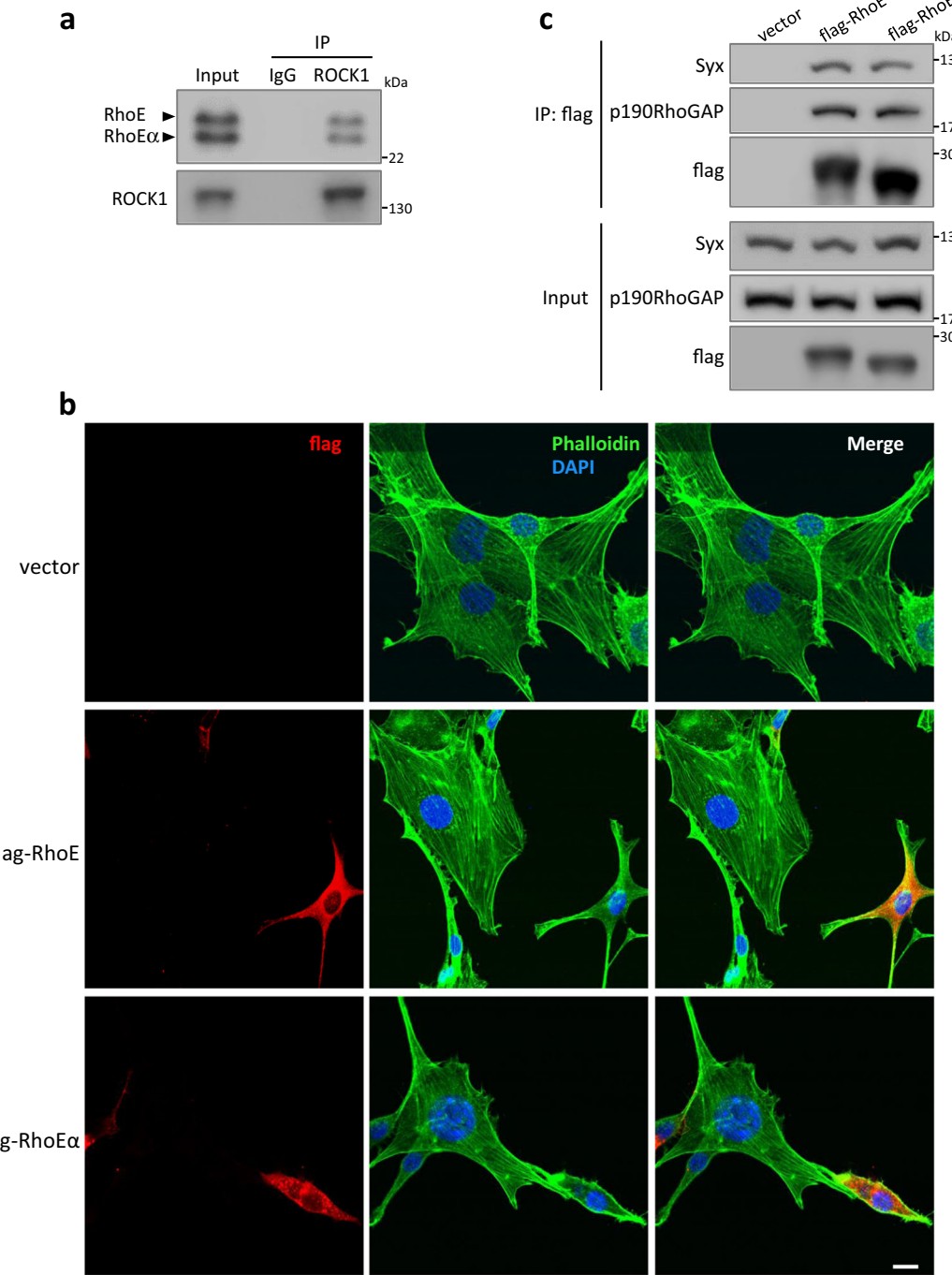

**Fig. 5 Functional similarity between RhoE and RhoEα. a** Co-IP assay detected binding of ROCK1 to both RhoE and RhoEα in HEK293 cells. **b** Confocal images of NIH-3T3 cells. Cells were transfected with the indicated plasmids and fixed 24 h after transfection. Flag-tagged proteins were stained with anti-flag antibody (red). Cytoskeleton actins were stained by phalloidin (green). Cell nuclei was stained by DAPI (blue). Scale bar: 10 μm. **c** Co-IP assay detected binding of both RhoE and RhoEα to Syx and p190RhoGAP. HEK293 cells were transfected with indicated plasmids. Twenty four hours after transfection, cell lysates were used for Co-IP assay with anti-flag antibody.

applications in a broad spectrum of cell biology and cancer research. We then reintroduced RhoE and RhoEα individually into the *RhoE* null cell lines, by transient transfection followed by RNA-seq analysis 24 h after transient transfection. RNA-seq data from *RhoE* null cells were used as a control for the analysis. The efficacy and fidelity of expression of RhoE and RhoEα were validated by western blot. RhoEα protein was exclusively expressed by flag-RhoEα, but not flag-RhoE expression vector (Supplementary Figs. 3a and 4a).

We found that introduction of RhoE and RhoEα resulted in obvious changes in 920 and 898 genes (more than twofold), respectively, in the HEK 293 cell line (Supplementary Fig. 3a, b). Among them, about two-third of gene expressions were uniquely induced by the introduction of two isoforms (583 and 561, respectively). Ingenuity pathway analysis (IPA) was performed in the two unique sets of genes and we observed a distinct difference in the general canonical pathways (Supplementary Fig. 3c for RhoE and Supplementary Fig. 3d for RhoEα). Of note, RhoE was

closely involved in VEGF-related pathways, consistent with our previous report on the regulatory role of RhoE in responsive cardiac angiogenesis[16]. RhoEα was associated with metabolism pathways, such as tyrosine degradation, noradrenaline and adrenaline degradation, and tryptophan degradation to 2-amino-3-carboxymuconate semialdehyde, etc. The differential expression levels among genes between RhoE and RhoEα group were validated by RT-qPCR, and phospholipase A2 group 10 (PLA2G10) and homogentisate 1,2-dioxygenase (HGD) were chosen for the assessment (right panels in Supplementary Fig. 3c, d).

Using the same strategy, we analyzed the gene expression profiles in the HeLa cell line after the introduction of the two RhoE isoforms (Supplement Fig. 4a). The reexpression of RhoE and RhoEα led to 1485 and 1405 gene transcript changes in more than twofold, respectively. The expression pattern induced by the two isoforms was similar to the observation detected in the HEK 293 cell line. Among the 1485 genes altered by RhoE, only 580 genes were overlapped with the genes induced by RhoEα (Supplementary Fig. 4a, b). Moreover, 905 and 825 genes were specifically induced by RhoE and RhoEα, respectively. IPA analysis of these specifically induced genes revealed cross talk, as well as distinct pathways (Supplementary Fig. 4c, d). Both RhoE and RhoEα were associated with tyrosine degradation, retinoid X receptor activation, eicosanoid signaling, and cAMP-mediated signaling. Meanwhile, RhoE showed a strong connection to TREM1 signaling and glutamate receptor signaling, while RhoEα seemed associated with protein citrullination. The differential expression levels among genes between RhoE and RhoEα group were validated by RT-qPCR and caspase 5 (Casp5) and leukotriene-B(4) omega-hydroxylase 1 (CYP4F2) were chosen for the assessment (right panels in Supplementary Fig. 4c, d).

## Discussion

RhoE is a small GTPase universally expressed in tissues and organs, and has shown diverse biological functions[4,36]. Animal studies have demonstrated that general RhoE gene deletion in mouse led to early embryonic lethality due to fetal arrhythmias[14]. Abnormal neuron development, hydrocephalus, myocardial apoptosis and inflammation, and tumor growth and metastasis have also been linked to the downregulation of RhoE[15–18,24,36,37]. Despite such diverse and critical functions of RhoE, regulation of RhoE expression and activity remains elusive and most studies have been focused on the regulatory effects of RhoE protein by PTMs, such as phosphorylation and farnesylation[38,39] and microRNAs[40–44]. In the present study, we uncover RhoEα as a new RhoE isoform, and demonstrate its existence in both tissues and multiple cell lines. RhoEα and RhoE share a similar expression pattern, subcellular distribution, and binding capacity to downstream effector ROCK1, p190RhoGAP, and Syx in physiological conditions. We further find that RhoEα protein exhibits a longer half-life and divergent regulatory pathways compared to RhoE. The similar and differential functions of RhoEα may explain how the fundamental and diverse functions are achieved by this small GTPase. The discovery expands the scope of small GTPases and there may exist isoforms of other GTPases in such a manner.

Protein synthesis, or translation of mRNA into amino acids, represents the final stage in the flow of genetic information. More than a simple assembly platform, recent studies have uncovered the critical regulatory role of the translation process both in manipulating protein expression and expanding proteome diversity[45,46]. Growing evidence shows that the translation machinery has great hidden coding potential, and may selectively initiate translation upstream and/or downstream of the annotated

coding sequence[47,48]. Of note, one screening study in global translation initiation has reported 6991 possible aTIS in 4961 human genes and 9973 aTIS in 5668 mouse genes[31]. Among these predicted aTIS, there is one at the 5′ UTR of RhoE in the database[31]. Interestingly, our study has excluded this presumed aTIS, since the insertion of a stop codon at the end of RhoE 5′ UTR has shown no effect on either RhoE or RhoEα expression. Instead, we have identified ATG[46], an aTIS downstream of the canonical translational start site ATG[1], responsible for the expression of RhoEα. Individual mutation of ATG[1] and ATG[46] at the RhoE coding sequence diminishes the expression of RhoE and RhoEα, respectively, suggesting that continual identification and validation of these new translational products by bench experiments are critically important.

PTM, i.e., phosphorylation, can cause unpredictable changes to protein electrophoretic mobility[49,50]. Madigan et al. observed phosphorylation of RhoE upon PKC agonist stimulation, which led to an electrophoretic mobility shift of exogenously expressed RhoE[33]. Though band shift of RhoE occurs after phosphorylation and other possible types of PTM, the extra RhoE band identified in our study is clearly derived from the alternative translation. Protein expression analysis with the ATG[1] and ATG[46] point mutations demonstrates that the upper band is RhoE, while the short isoform RhoEα is derived from alternative translational site ATG[46]. Obviously, as the function and localization of RhoE are tightly regulated by phosphorylation and farnesylation, correlated PTM studies of RhoEα should further expand our knowledge about this new RhoE isoform.

RhoE localizes in both cell cytosol and membrane to mediate its regulatory roles[6,7]. Our fluorescent images show that RhoEα exhibits a similar localization pattern with RhoE in normal cell culture conditions, indicating that the first 15 amino acids in the N-terminus of RhoE may not contribute to its subcellular localization. The result is consistent with the protein structure prediction (Supplementary Fig. 5) and the crystal structure study of RhoE[51], which showed the absence of secondary structures within the first 21 amino acids in RhoE's N-terminus. One interesting question for future study is whether RhoE and RhoEα coordinate or counteract each other in a heterodimer manner, which is commonly observed among Ras family members[52,53]. Meanwhile, given the fact of the dynamic movement of RhoE between cytosol and cell membrane during important biological processes, such as membrane blebbing, cell rounding, and migration[21,54,55], understanding the role of RhoEα in these biological processes becomes equally important.

Another interesting observation is that expression levels of the two proteins are not always equal. A higher expression level of RhoEα is often detected. In this study, we have demonstrated that RhoEα is more stable with a longer half-life than RhoE, which partially explained the unequal expression levels of the two isoforms. We also ruled out the possibility of RhoEα-mediated degradation of RhoE. Meanwhile, we realized that a bioinformatic analysis revealed a strong Kozak sequence around ATG[46] compared to the one near ATG[1], which may contribute to the varying levels of the two proteins. Goh et al. reported that RhoE can be stabilized by its effectors, such as Syx and p190RhoGAP[56]. We repeated their result on RhoEα and further found that RhoEα was also able to interact with the two effectors, with similar binding affinities compared to RhoE. The increased protein stability of RhoEα may indicate the possibility of uncovered RhoEα-specific effectors that could enhance RhoEα stability or inhibit its degradation, which is an interesting topic for future study.

While the major discovery of the study was the identification and characterization of a new isoform RhoEα, several basic studies on the function of RhoEα were explored. The results indicated functional similarity between the two isoforms, which

included physical interactions with ROCK1, p190RhoGAP, Syx, and the inhibitory effect on actin assembly.

Finally, we compared the differential transcriptome profiles and the associated signaling pathways between RhoE and RhoEα in two cell lines. The data indicate that one-third of genes are commonly regulated by the two proteins, and the remaining are differentially regulated by RhoE and RhoEα individually. The IPA revealed some consistent, as well as distinct signaling pathways associated with the two proteins. For an example, VEGF, inflammation, and cAMP-mediated signaling have been reported previously[14,16,17].

We would like to indicate that this part of the study is in its primary stage and multiple factors, including different backgrounds of the two cell lines can potentially impact the outcome. Interpretation of the result should be cautious and the result serves as the basis for future study. Further exploration of differential functions by the two isoforms is necessary and profound.

In summary, this study has defined a new isoform of small GTPase, RhoEα. The distinct characteristics of RhoEα and its potential functions have been investigated. The newly identified RhoEα provides a new layer of regulatory complexity to the RhoE-mediated regulatory program, and an alternative angle to understand the versatile functions of this small GTPase.

## Methods

**Animals**. Tissues were collected from male C57BL/6 mice at the age of 10 weeks. *RhoE* null MEF cells were isolated from *RhoE* null (*RhoE*$^{-/-}$) mice[14]. All animal experiments were approved by the Institutional Animal Care and Use Committee of Texas A&M University College of Medicine, Institute of Biosciences and Technology.

**siRNAs, plasmids, oligos, and antibodies**. Nontargeting siRNA (D-001810-01-05), human SMARTpool *RhoE* siRNA (L-007794-00-0005), and mouse SMARTpool *RhoE* siRNA (L-064484-01-00005) were from GE Dharmacon. Restriction enzyme-based cloning was used for plasmid construction. For 5′ UTR-RhoE and 5′ UTR-RhoE-flag constructs, 5′ UTR and coding sequence of RhoE cDNA (ENST00000263895.8) was amplified by PCR from HeLa cell cDNA library, and inserted into pcDNA3.1 and CMV-COOH-3xflag vector. For flag-RhoE and GFP-RhoE constructs, human RhoE coding sequence was inserted into CMV-NH2-3xflag and pEGFP-C3 vector. For flag-RhoEα and mCherry-RhoEα, human RhoE coding region starting from ATG$^{46}$ was inserted into CMV-NH2-3xflag and pmCherry-C1 vector. Plasmids with point mutation were constructed using the Q5 site-directed mutagenesis kit (NEB, E0554S). Oligos used for RT-qPCR included: PLA2G10: GGTTGCTTTTGTGGCTTGGGAG and GATTGACGCACTGCCAG-GAGTA; HGD: CATCTTGGAGGTCTATGGTGTCC and GACCGTGTAACCA CCTGGTACT; Casp5: ACAACCGCAACTGCCTCAGTCT and GAATCTG CCTCCAGGTTCTCAG; and CYP4F2: GACAGCCATTGTCAGGAGAAACC and TGCAGGAGGATCTCATGGTGTC. The antibodies used in the study included: RhoE (EMD Millipore, 05-723, LOT: 2802018), GAPDH (Cell Signaling Technology, #5174), flag (Sigma Aldrich, F1804-1MG), ROCK1 (Santa Cruz Biotechnology, sc-5560), Syx (ThermoFisher Scientific, PA5-62010), p190RhoGAP (Santa Cruz Biotechnology, sc-393241), anti-mouse IgG HRP-linked secondary antibody (Cell Signaling Technology, #7076), anti-rabbit IgG HRP-linked secondary antibody (Cell Signaling Technology, #7074), conformation-specific anti-mouse IgG HRP-linked secondary antibody (Abcam, ab131368), and conformation-specific anti-rabbit IgG HRP-linked secondary antibody (Cell Signaling Technology, #5127).

**Mass spectrometry**. HEK 293 cell lysate was subjected to anti-RhoE antibody pull-down followed by SDS–PAGE gel separation. The gel was stained with 0.1% Coomassie brilliant blue R250 and two bands ~27 kDa were excised. The samples were digested by trypsin. The mass spectrometry analysis was performed by the Mass Spectrometry Facility at the University of Texas Medical Branch at Galveston using SCIEX TOF/TOF 5800 system.

**CRISPR-mediated genome editing**. The CRISPR-mediated genomic deletion was performed as described previously[57]. For *RhoE* deletion in HeLa and HEK 293 cells, guide RNA (GGGCGGACATTGTCATAGTA) specifically targeting on *RhoE* exon 4 was cloned into pSpCas9(BB)-2A-Puro (PX459) V2.0 (Addgene, #62988). Cells were transfected with the fused plasmid by lipofectamine 2000. Two days after transfection, the cells were treated with puromycin at 2.5 μg/ml for 3 days. Puromycin-resistant cells were harvested and further seeded into a 96-well plate,

using a serial dilution for single-cell clones. Cell clones were validated by immunoblot. For CRISPR-mediated activation of *RhoE* promoter, HeLa cells were transfected with the Cas9-SAM components, including dCas9-VP64-GFP (Addgene, #61422), MS2-P65-HSF1-GFP (Addgene, #61423), and sgRNA(MS2) cloning vector (Addgene, #61424) fused with guide RNA (ATCTGCCTCCTC CCCTTTTA) targeting on human *RhoE* promoter region. For CRISPR-mediated genomic point mutation, guide RNA (TTTGCTGGATAATTTCTGGC) targeting on *RhoE* exon 2 was cloned into pSpCas9(BB)-2A-Puro (PX459) V2.0. The single-stranded donor oligonucleotide (ssODN) designed as repair template to create ATG$^1$ deletion mutation is CACACTGACTGTCTCCCACCACAACTATCT TGC ATTTCACGTTCTGATTAGGATCCATGATAGATTTGCTGGATAATTTC TGGCTTGCTCTTCTCTCCTTTGATGTTGCCTTATTTTCTCTTGGAACAG GAATTTTCTCTTAAGAAG. The ssODN as repair template to create ATG$^{46}$ deletion mutation is CACACTGACTGTCT CCCACCACAACTATCTTGCATTT CACGTTCTGATTAGGATCGATAGATTTGCTGGATAATTTCTGGCTTGCT CTTCTCTCCTTCATTGATGTTGCCTTATTTTCTCTTGGAACAGGAATTTT CTCTTAAGAAG. HeLa cells were co-transfected with the sgRNA/Cas9 containing plasmid and the ssODN. Two days after transfection, the cells were treated with puromycin at 2.5 μg/ml for three days. Puromycin-resistant cells were subjected to immunoblot assay.

**Immunofluorescent staining and fluorescent imaging**. NIH-3T3 cells were cultured in a 35-mm culture dish (MatTek, P35G-1.5-14-C) for overnight and then transfected with three expression vectors control, flag-RhoE and flag-RhoEα individually. Twenty four hours after the transfection, cells were washed and fixed by 4% paraformaldehyde for 30 min. The cells were probed by the anti-flag antibody (Sigma Aldrich, F1804-1MG) followed by the second antibody, Alexa Fluor 594 goat anti-mouse IgG (ThermoFisher Scientific, A-11005). Alexa Fluor 488 Phalloidin (ThermoFisher Scientific, A12379) was used for cytoskeleton actin staining. For fluorescent imaging analysis, HeLa cells were co-transfected with GFP-RhoE and mCherry-RhoEα plasmids. Imaging was taken by Nikon Confocal A1 laser microscope.

**Cycloheximide chase assay**. The degradation of RhoE and RhoEα was analyzed by CHX chase assay. HeLa cells were seeded in a six-well plate, incubated overnight, and subsequently treated with 20 μg/ml CHX for 0, 3, and 6 h, respectively. Cell lysates were then collected and subjected to immunoblotting assay for detection of RhoE and RhoEα levels. Immunoblots were qualified by densitometry of the film using ImageJ software. Three independent experiments were performed, and the data were analyzed by the associated statistical analysis.

**RNA sequencing**. RNA was isolated from HEK 293 cells and HeLa cells using TRIzol reagent. For each sample, a total of 10 μg RNA was used for mRNA purification and cDNA library preparation according to the Ultra Directional RNA Library Prep Kit for Illumina (NEB, E7420S). The quality control of the library quality control was assessed by Qubit (Thermo Fisher) and qPCR. The libraries were sequenced using an Illumina Novaseq 6000 platform with paired-end 150 bp strategy. The quality of raw RNA-sequencing data was validated by the FastQC software (www.encodeproject.org/software/fastqc/). The sequencing reads were aligned to hg38 using Tophat and the reads mapping to each of the 56,269 genes/locations were counted using HTSeq. R package DESeq was used to normalize and generate the variance of stabilizing transformation data. The fold changes were calculated for differential gene expression analysis. IPA was used to analyze the differentially expressed genes with |fold change| ≥ 2. The raw RNA-seq data were deposited in the Gene Expression Omnibus (GEO) with accession number GSE132718.

**Statistics and reproducibility**. For two-group comparisons, unpaired student's *t* test was used. A value of *p* < 0.05 was considered statistically significant. For protein stability assay, the experiment was independently replicated three times and the results were qualified together. Experiments and data assessment were conducted double blinded by investigators.

**Reporting summary**. Further information on research design is available in the Nature Research Reporting Summary linked to this article.

## Data availability

The data supporting the findings of this study are available within the paper and its' Supplementary information. Uncropped images of the coomassie blue-stained gel and immunoblots are presented in Supplementary Fig. 6 and Supplementary Fig. 7. The raw RNA-seq data were deposited in the GEO with accession number GSE132718. Other Source data related to the study are available from the corresponding author upon reasonable request.

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

## Acknowledgements

This study was supported by American Heart Association Transformational Project Award 19TPA34880011 (J.C.) and the NIH-NHLBI R01HL141215 (J.C.).

## Author contributions

Y.D. and J.C. designed the experiments; Y.D. conducted most of the experiments; Y.D. and W. L. collected and analyzed the data. X.Y. generated RhoE$^{-/-}$ MEF cells. W.M. and J.W. processed the RNA-seq data and performed bioinformatic analysis. Y.D. and J.C. wrote the manuscript. All of the authors edited the manuscript and approved the final manuscript.

## Competing interests

The authors declare no competing interests.
