## [Peer Review File · Communications Biology]

Reviewers' comments:

Reviewer #1 (Remarks to the Author):

This manuscript reports the identification and characterization of a new isoform of RhoE, a member of the Rho GTPase family.

Even if this paper provides interesting results about the identification of a new isoform of RhoE, it is unfortunate that the characterization part fails to convey the message properly. Further experiments are required to strengthen the statements.

Here are my specific comments.

1. Within the Rho family of GTPases, Rac1b was described as an alternatively spliced isoform of Rac1. Thus RhoEa will not be the first isoform reported. This should be corrected in the manuscript and paper about Rac1b should be cited.
2. The first paragraph of the result section is misleading. The results shown on figure 1 do not necessary demonstrate that the two bands are RhoE isoforms, it could also be a post-translational modification that leads to a mobility shift on gel.
3. On figure 4A, it is striking to see a complete overlap between both stainings. For Nter-tagged versions of RhoE, either GFP-RhoE or Flag-RhoE, is RhoEa expressed from the RhoE construct? It is important to show an immunoblot analysis with anti-GFP/anti-Flag and anti-RhoE antibodies for Figure 4 and Figure 5.
4. To be sure that RhoEa is not translated from RhoE cDNA, the used of the RhoE AAA46 mutant would be useful. To compare the subcellular localization of both isoforms, a co-staining between GFP-RhoE AAA46 and mCherry-RhoEa is necessary. Alternatively an antibody raised against the 15 N-terminal amino acids to localize only RhoE and not RhoEa would be useful.
5. Without any further validation, the transcriptomic analyses are not very informative and seem cell -type dependent. Are results of both cell lines very different? What is the overlap between HeLa and HEK293 transcriptomic profiles? Again an immunoblot analysis with anti-Flag and anti-RhoE antibodies is important to show to know if RhoEa expressed upon flag-RhoE construct transfection. If so, transcriptomic analyses should be performed with the mutant RhoE AAA46
6. RhoE is also known as Rnd3 and as the name dictates, Rnd3 protein was associated with making a cell "round" by antagonizing RhoA activity. Thus, the impact of the expression of RhoE AAA46 (encoding only RhoE) versus RhoEa on stress fibers, focal adhesions and/or RhoA activity is missing.
7. Comparison of the impact of siRNAs targeting both isoforms and siRNAs targeting only RhoE would be very informative.

Reviewer #2 (Remarks to the Author):

The manuscript, "Identification and characterization of a new isoform of small GTPase RhoE," by Dai. al. reports a novel isoform of RhoE, herein named RhoEa, which may have distinctive biological functions from those of RhoE.

Major Concerns:

- 1) While the observation is indeed an interesting new finding to the field of RhoGTPases, the major concern is that the current manuscript is largely descriptive and lacks the sophisticated mechanistic studies to determine functional and translational consequences in determining the biological role of RhoEa and/or if it is indeed different, redundant or compensatory to the role of RhoE. Further analysis of the RhoEa isoform in a mammalian model system is required and to determine functional, biological, and translational consequences of the RhoEa isoform.
- 2) Are there known mutations in RhoE or RhoEa that have been identified from genetic databases? How does the expression of one affect the other in terms of functional correlation?

- 3) Fig 3C and 3D: Why does KO or CRISPR of the RhoE α isoform affect the stable expression of RhoE? Is RhoE degraded in the absence of RhoE α ? These data shown suggest that RhoE α (ATG46) is required for RhoE (ATG1) expression and function. Further studies are required to parse out these details.
- 4) Fig 4: Additional cells should be indicated in the IF panel A. What is transfection efficiency here?
- 5) Are there differential effects on postranlational modifications on RhoE or RhoE α , ie phosphorylation, farnesylation, etc? What are the functional consequences of the new isoform vs RhoE?
- 6) Fig 5 and 6: Are there functional domains lost in the first 46 aa in RhoE α vs RhoE that can potentially account for the differential effects in downstream genes? How exactly are the effects on inflammation differentially regulated by the two isoforms? Could this be an effect of the differing expression levels of the two isoforms? What about differences in the transfection efficiency? This may simply be a kinetics shift (from increased or decreased expression of each isoform) that falsely indicates differential gene patterning between RhoE vs RhoE α . Evidence to the contrary would be needed and could be novel if proven.
- 7) Does RhoE and/or RhoE α form homo or heterodimers in vivo? Which then would be rate limiting? Could they compete for downstream effector signaling?

Minor:

- 1) Panel 3D should be labeled.
- 2) Fig 4: Additional cells (lower magnification) should be indicated to show the level of expression in the dish, particularly since this is transient.
- 3) Data showing the effects on expression of RhoE or RhoE α from the predicted a TIS at the 5'UTR of RhoE should be shown.

Reviewer #3 (Remarks to the Author):

In this manuscript, the authors describe RhoE α , a shorter RhoE isoform. They provide compelling evidence that this isoform results from the same gene by alternative translation initiation at the downstream ATG codon 46. Expression of a RhoE α cDNA suggests that the protein did not differ in subcellular localization but has increased protein stability compared to RhoE. Finally, they expressed RhoE or RhoE α individually in cells and found differential transcriptome changes using RNA-seq. They conclude that both isoforms determine distinct molecular signaling profiles. While the data on the origin of the RhoE α isoform are solid and well-supported by experimental data, the analysis of potential differences is at a preliminary stage.

Main concerns:

- 1) page 3 last paragraph, misleading sentence: "To our knowledge, RhoE α is the first isoform of the members in the Rho GTPase family."
According to the review cited as reference 2, the Rho GTPase family belongs to the RAS superfamily and is subdivided into the following subfamilies: Rnd, Rho-like, Rac-like, Cdc42-like and RhoBTB. Isoforms through alternative splicing have been described for RAC1 and Cdc42, so it is not correct that RhoE α is the "first isoform of the members in the Rho GTPase family". Rather, it is the first member in the Rnd-subfamily of Rho GTPases.
- 2) The manuscript lacks a comparison and discussion of the Kozak consensus sequence around ATG1 and ATG46. The RhoE exon 1 sequence, which is available from the Ensemble database, reveals that only ATG46 has a strong Kozak motif.
 - a) this leads to preferential translation initiation at ATG46 and needs to be addressed in the manuscript. In addition to "RhoE α being more stable with a longer half-life than RhoE", as the authors discuss, the preferential translation would be a major factor explaining the observed higher expression level of RhoE α ;

b) the sequence comparison shown in Fig 2 should include an additional 5 nucleotides 5' to ATG1 so that this Kozak effect can be recognized by the reader;

c) Fig 3 C and D seem to indicate that in the absence of ATG46, the full-length RhoE form (that starts at ATG1) is drastically reduced. Is this a matter of stability of RhoE protein? Can this be rescued by co-transfecting construct 3 and 4 from Fig 3C (i.e. RhoE and RhoE α) into the RhoE-KO HeLa cells?

3) Concerning the functional divergence between RhoE and RhoE α

a) This part of the manuscript lacks any experimental validation of the differentially expressed genes, which were identified by RNAseq. Quantitative PCR needs to be done for at least some candidate genes, otherwise these experiments are not very meaningful. In particular highlighting pathways that are very distant from the cell types used, including autoimmune response or liver cell metabolism, make the value of this analysis questionable. Bioinformatic filtering of sequence data can yield very different outcomes and the mere listing of functional categories as presented in this part of the manuscript is neither informative nor helpful for the isoform characterization.

b) concerning the relative contribution of this RNAseq data interpretation to the remaining manuscript, this part is of excessive length. In addition, the lack of informativity of Figs 5 and 6 would suggest to present them as supplemental material and include a new Figure 5 that presents some experimental validation of suspected candidate genes.

c) page 9, first sentence (also page 3), the authors state that they compared the molecular signaling profiles of the two proteins; however, they analyzed their gene expression profiles, which is not the same. This should be corrected.

d) It might be interesting to add a simple anti-flag co-immunoprecipitation experiment comparing both isoforms individually with respect to the described interaction with effectors such as Syx and p190RhoGAP45.

minor corrections:

5) page 3 last paragraph: "and this site is located at the downstream of the classical RhoE translation start site" should be "and this site is located downstream of the classical RhoE translation start site."

6) page 5, paragraphs on Confocal imaging and on Cycloheximide (CHX) chase assay: the sentence "HeLa cells were seeded in a 6-well plate for overnight and subsequently (..)" is not accurate and should be corrected, e.g. to "HeLa cells were seeded in a 6-well plate, incubated overnight and subsequently (..)"

7) Figure 1F: although the MS-identified peptides unequivocally belong to Rho E, the two drawn lines pointing to the excised bands do not seem to indicate the bands at 25 and 27 kDa; please check carefully, especially since the antibodies do not seem to be very specific or efficient in RhoE immunoprecipitation

8) page 7, second paragraph: in the sentence "Alternative translation initiation and alternative splicing from a single gene are the two major mechanisms resulting in a generation of protein isoform in most circumstances^{28, 29}", the cited references do not include work from the alternative splicing field. Second, it would be advisable to refer at this stage that another possibility for the observed isoform band pattern could be a post-translational modification such as phosphorylation that frequently causes a shift in electrophoretic mobility.

9) page 7 and Figure 2: The authors argue that the transfected RhoE cDNA also yields the expression of two bands in cells, thus indicating alternative translation initiation. It is important to state more clearly how this cDNA construct was cloned and what sequences from its 5' or 3' untranslated regions were included. The corresponding sentence in the materials and method section is currently: "RhoE cDNA (ENST00000263895.8) was amplified by PCR from HeLa cDNA and inserted into pcDNA3.1 or COOH-3xflag vector." but Fig 2 indicates the presence of 5-utr sequence.

10) page 7: grammar error: "and the alternative translation start site should be located at the downstream of ATG1. should be: "and the alternative translation start site should be located downstream of ATG1".

11) page 8: style: "Collectively, these data provide evidence that RhoE α is a new isoform of RhoE with 15 amino acids shorter compared to RhoE." this would be better as : "...that the new isoform RhoE α is 15 amino acids shorter compared to RhoE."

We thank the reviewers and editor for the constructive and insightful comments, which greatly improved the quality of the manuscript. The following is the responsive summarization pertaining to the reviewer's specific comments point-by-point.

Reviewer #1:

This manuscript reports the identification and characterization of a new isoform of RhoE, a member of the Rho GTPase family.

Even if this paper provides interesting results about the identification of a new isoform of RhoE, it is unfortunate that the characterization part fails to convey the message properly. Further experiments are required to strengthen the statements.

Here are my specific comments.

1. Within the Rho family of GTPases, Rac1b was described as an alternatively spliced isoform of Rac1. Thus RhoE α will not be the first isoform reported. This should be corrected in the manuscript and paper about Rac1b should be cited.

Response: We thank the reviewer for pointing out the error. The manuscript was revised accordingly. Two alternative splicing references (ref. 25 and 26 in manuscript) were cited^{1, 2}.

2. The first paragraph of the result section is misleading. The results shown on figure 1 do not necessary demonstrate that the two bands are RhoE isoforms, it could also be a post-translational modification that leads to a mobility shift on gel.

Response: That's a good point. We revised the text accordingly. The possibility of a mobility shift by a post-translational modification was also discussed in the result of figure 3.

3. On figure 4A, it is striking to see a complete overlap between both stainings. For Nter-tagged versions of RhoE, either GFP-RhoE or Flag-RhoE, is RhoE α expressed from the RhoE construct? It is important to show an immunoblot analysis with anti-GFP/anti-Flag and anti-RhoE antibodies for Figure 4 and Figure 5.

Response: As suggested, we performed Western blot showing that RhoE α cannot be expressed by N-terminal-tagged-RhoE constructs (Figure 4A, Supplement Figure 3A and 4A).

Figure 4A. Immunoblot showed only RhoE but not RhoE α expression from the GFP-RhoE expression plasmid.

Supplement Figure 3A and 4A. Immunoblot analysis for flag-RhoE and flag-RhoE α expression in HEK 293 cells (S3A) and HeLa cells (S4A).

To consolidate the result and give a whole picture of localization of two isoforms, we added new fluorescent images exhibiting the result in multiple cells (Figure 4B, top panel).

Figure 4B. Confocal images exhibited subcellular localizations of GFP-RhoE and mCherry-RhoE α in the transfected HeLa cells.

4. To be sure that RhoE α is not translated from RhoE cDNA, the used of the RhoE AAA46 mutant would be useful. To compare the subcellular localization of both isoforms, a co-staining between GFP-RhoE AAA46 and mCherry-RhoE α is necessary. Alternatively an antibody raised against the 15 N-terminal amino acids to localize only RhoE and not RhoE α would be useful.

Response: Yes, we took reviewer's suggestion and performed a fluorescent imaging experiment with co-expressing RhoE (green) and RhoE α (red) by GFP-RhoE ATG⁴⁶ > AAA construct and mCherry-RhoE α construct, respectively. The result is consistent and supportive for the conclusion. The new data were presented as Supplement Fig. 2.

Supplement Figure 2. Confocal images exhibited subcellular localizations of GFP-RhoE with ATG⁴⁶ to AAA mutation and mCherry-RhoE α in HeLa cells.

5. Without any further validation, the transcriptomic analyses are not very informative and seem cell-type dependent. Are results of both cell lines very different? What is the overlap between HeLa and HEK293 transcriptomic profiles? Again an immunoblot analysis with anti-Flag and anti-RhoE antibodies is important to show to know if RhoE α expressed upon flag-RhoE construct transfection. If so, transcriptomic analyses should be performed with the mutant RhoE AAA46.

Response: We took the reviewer's suggestion and validated the efficacy and fidelity of expression of RhoE and RhoE α by Western blot. RhoE α protein was exclusively expressed by flag-RhoE α but not flag-RhoE expression vector. The new data were included in Supplement Fig. 3A and Supplement Fig. 4A.

Supplement Figure 3A and 4A. Immunoblot analysis for flag-RhoE and flag-RhoE α expression in HEK 293 cells (S3A) and HeLa cells (S4A).

While the major discovery of the study is the identification and characterization of a new isoform RhoE α , we agree that the study of transcriptomic analysis is limited. To address the concern, three major revisions were added: 1) organ- and/or tissue-specific signaling was filtered out in bioinformatic analysis; 2) RNA-seq data were validated by qPCR; 3) limitation of the study was discussed. Finally, Fig. 5 and 6 were moved to supplemental section.

6. RhoE is also known as Rnd3 and as the name dictates, Rnd3 protein was associated with making a cell “round” by antagonizing RhoA activity. Thus, the impact of the expression of RhoE AAA46 (encoding only RhoE) versus RhoE α on stress fibers, focal adhesions and/or RhoA activity is missing.

Response: That is an excellent point and we shared the same curiosity with reviewer. As suggested, we examined the effect of RhoE α on cell actin assembly in NIH-3T3 cells. Disrupted actin fibers were obviously exhibited in both RhoE α and RhoE transfected cells. The new data were added in Fig. 5B.

Figure 5B. Confocal images in NIH-3T3 cells transfected with the indicated plasmids. F-actin was stained with phalloidin and nucleus was stained with DAPI.

7. Comparison of the impact of siRNAs targeting both isoforms and siRNAs targeting only RhoE would be very informative.

Response: We understand reviewer’s concern and share with the opinion. However, current siRNAs targeting technology could not reach what the reviewer suggested. Both ATG¹ and ATG⁴⁶ are located at the same exon and the two isoforms are translated from the same mRNA with different translation start site. Because there is only one identical mRNA, siRNAs targeting for RhoE will also lead to a degradation of RhoE α mRNA. Therefore, we have to generate a cell line with *RhoE* null background and then introduce RhoE and RhoE α individually to investigate differential transcriptome profiles of two isoforms.

Reviewer #2 (Remarks to the Author):

The manuscript, "Identification and characterization of a new isoform of small GTPase RhoE," by Dai. al. reports a novel isoform of RhoE, herein named RhoE α , which may have distinctive biological functions from those of RhoE.

Major Concerns:

1) While the observation is indeed an interesting new finding to the field of RhoGTPases, the major concern is that the current manuscript is largely descriptive and lacks the sophisticated mechanistic studies to determine functional and translational consequences in determining the biological role of RhoE α and/or if it is indeed different, redundant or compensatory to the role of RhoE. Further analysis of the RhoE α isoform in a mammalian model system is required and to determine functional, biological, and translational consequences of the RhoE α isoform.

Response:

We agree with the reviewer that the investigation of functional, mechanistic and translational consequences of this newly identified RhoE α including animal studies is profoundly important. While great efforts have been made for understanding of RhoE in past two decades since the discovery of RhoE in 1996, many of "functional and translational consequences" of RhoE still remain to be explored. The same efforts should be made for this new RhoE α in future studies.

Current study demonstrated a novel RhoE isoform RhoE α , explored its basic biological characteristics, functions and the transcriptome profiles. The findings lay a foundation and open a window for a wide range of future studies including those suggested by the reviewer.

2) Are there known mutations in RhoE or RhoE α that have been identified from genetic databases? How does the expression of one affect the other in terms of functional correlation?

Response: As suggested by the reviewer, we performed a computer search from Human Gene Mutation Database (HGMD, QIAGEN). No mutations in both *RhoE* and *RhoE α* were found. The result is interesting and may suggest that *RhoE* or *RhoE α* is so essential that any individual with mutations in *RhoE* or *RhoE α* may not be preserved or survived during evaluation.

The question, "How does the expression of one affect the other in terms of functional correlation", was addressed in the following response to the question 3 due to their similarity.

3) Fig 3C and 3D: Why does KO or CRISPR of the RhoE α isoform affect the stable expression of RhoE? Is RhoE degraded in the absence of RhoE α ? These data shown suggest that RhoE α (ATG46) is required for RhoE (ATG1) expression and function. Further studies are required to parse out these details.

Response: We agree that this is an interesting phenomenon. To rule out if there is any direct effect of RhoE α on the expression of RhoE or RhoE protein stability, we re-introduced RhoE α in the cells and assessed the level of RhoE protein by Western blot. We found that forced expression of RhoE α had a minimal impact on RhoE protein level (Fig. 4D).

Meanwhile, we performed a sequence analysis and found a potent Kozak sequence around ATG⁴⁶, while the ATG¹ had a less optimal Kozak sequence (Fig. 3E). Preferential translation initiation at ATG⁴⁶ could be another causative factor contributing to high level expression of RhoE α . Since two initiation sites ATG¹ and ATG⁴⁶ are not far away, loss of the integrity of

ATG⁴⁶ Kozak motif (e.g. ATG⁴⁶>AAA) may weaken the translation initiation complex formation at ATG¹ and consequently reduce RhoE expression as shown in Fig. 3C and 3D.

Furthermore, this strong Kozak sequence and ATG⁴⁶ are highly conserved among vertebrate species (Figure 3E), supporting a broad existence of RhoE α expression across species.

Collectively, the differences in the proteins' stability and preferential translation initiation may lead to a varying level of the two proteins.

Figure 4D. Forced expression of RhoE α had minimal impact on RhoE protein level.

Figure 3E. Comparison of Kozak sequences around human RhoE translation initiation sites. The strongest Kozak context promoting translation initiation contains a purine, preferably adenine A, 3 bp before ATG, and a guanine G following ATG.

4) Fig 4: Additional cells should be indicated in the IF panel A. What is transfection efficiency here?

Response: That's a good point. We added a new set fluorescent images shown in Figure 4B (top panel) as suggested. Transfection efficiency is good and solid to support the experiments. 0.5 μ g GFP-RhoE and 0.5 μ g mCherry-RhoE α vectors were co-transfected in 4x10⁵ HeLa cells in 30-mm dish.

Figure 4B. Confocal images exhibited subcellular localizations of GFP-RhoE and mCherry-RhoE α in the transfected HeLa cells.

5) Are there differential effects on postranlational modifications on RhoE or RhoEa, ie phosphorylation, farnesylation, etc? What are the functional consequences of the new isoform vs RhoE?

Response: We agree with the reviewer that post-translational modifications (PTM) of RhoE, such as phosphorylation and farnesylation, are important regulatory mechanisms³⁻⁵. The current study didn't explore the functional consequences of PTM in RhoE α partially due to the unavailability of site-specific modification antibodies.

6) Fig 5 and 6: Are there functional domains lost in the first 46 aa in RhoEa vs RhoE that can potentially account for the differential effects in downstream genes? How exactly are the effects on inflammation differentially regulated by the two isoforms? Could this be an effect of the differing expression levels of the two isoforms? What about differences in the transfection efficiency? This may simply be a kinetics shift (from increased or decreased expression of each isoform) that falsely indicates differential gene patterning between RhoE vs RhoEa. Evidence to the contrary would be needed and could be novel if proven.

Response: We would like politely to correct reviewer's typing error, which RhoE α misses the first 15 aa (amino acids) not 46 aa compared to RhoE.

According to RhoE crystal structure, no functional domain nor secondary structure element are suggested in this part of N-terminus⁶. Interestingly, like RhoE α , most of other Rho family proteins do not contain this 15 aa, such as Rnd2, RhoA, RhoB, RhoC, RhoD, Rif, Rac1-3, RhoG, and Cdc42.

As we addressed to the concern 1, current study focused on the discovery of RhoE α and explored its basic biological characteristics, functions and the transcriptome profiles. The findings lay a foundation and open a window for a wide range of future studies including inflammation. Meanwhile, our study raised many questions, hypotheses or models for future as indicated by the reviewer, demonstrating the importance of the study.

The differential transcriptome patterning was not due to transfection efficiency. The expression levels of RhoE and RhoE α by transient transfection were even and consistent as shown in the Supplement Figure 3A and Supplement Figure 4A.

Supplement Figure 3A and 4A. Immunoblot analysis for flag-RhoE and flag-RhoE α expression in HEK 293 cells (S3A) and HeLa cells (S4A).

7) Does RhoE and/or RhoEa form homo or heterodimers in vivo? Which then would be rate limiting? Could they compete for downstream effector signaling?

Response: This an interesting idea and hypothesis. Currently, no report shows RhoE forming a homo to our knowledge.

In this study, we observed a colocalization of two isoforms and demonstrated RhoE α 's binding capability to RhoE's effectors, ROCK1, p190RhoGAP and Syx. We also detected similar levels of RhoE and RhoE α by ROCK1 pulldown assay (Fig. 5A). These results may indicate possibility for the formation of homo or heterodimers of the two isoforms, however, further experiments are required for a definitive answer.

Figure 5A. Co-IP assay detected binding of ROCK1 to both RhoE and RhoE α in HEK293 cells.

Minor:

1) Panel 3D should be labeled.

Response: Yes, we revised accordingly.

2) Fig 4: Additional cells (lower magnification) should be indicated to show the level of expression in the dish, particularly since this is transient.

Response: Yes, new images containing multiple cells were added as shown in Fig. 4B.

Figure 4B. Confocal images exhibited subcellular localizations of GFP-RhoE and mCherry-RhoE α in the transfected HeLa cells.

3) Data showing the effects on expression of RhoE or RhoE α from the predicted aTIS at the 5'UTR of RhoE should be shown.

Response: Yes, we added the new data as Supplement Figure 1.

Supplement Figure 1. Predicted aTIS is not responsible for new RhoE isoform expression.

Reviewer #3 (Remarks to the Author):

In this manuscript, the authors describe RhoE α , a shorter RhoE isoform. They provide compelling evidence that this isoform results from the same gene by alternative translation initiation at the downstream ATG codon 46. Expression of a RhoE α cDNA suggests that the protein did not differ in subcellular localization but has increased protein stability compared to RhoE. Finally, they expressed RhoE or RhoE α individually in cells and found differential transcriptome changes using RNA-seq. They conclude that both isoforms determine distinct molecular signaling profiles.

While the data on the origin of the RhoE α isoform are solid and well-supported by experimental data, the analysis of potential differences is at a preliminary stage.

Main concerns:

1) page 3 last paragraph, misleading sentence: "To our knowledge, RhoE α is the first isoform of the members in the Rho GTPase family."

According to the review cited as reference 2, the Rho GTPase family belongs to the RAS superfamily and is subdivided into the following subfamilies: Rnd, Rho-like, Rac-like, Cdc42-like and RhoBTB. Isoforms through alternative splicing have been described for RAC1 and Cdc42, so it is not correct that RhoE α is the "first isoform of the members in the Rho GTPase family". Rather, it is the first member in the Rnd-subfamily of Rho GTPases.

Response: We greatly appreciate the review for pointing out the error. The statements were corrected and two original alternative splicing studies in Rho GTPase^{1, 2} were cited as reference 25 and 26 in manuscript.

2) The manuscript lacks a comparison and discussion of the Kozak consensus sequence around ATG1 and ATG46. The RhoE exon 1 sequence, which is available from the Ensemble database, reveals that only ATG46 has a strong Kozak motif.

Response: That's a fabulous point! We performed an analysis and found a potent Kozak sequence around ATG⁴⁶, while the ATG¹ had a less optimal Kozak sequence (Fig. 3E). Furthermore, this strong Kozak sequence along with ATG⁴⁶ are highly conserved among vertebrate species (Figure 3F), supporting a broad existence of RhoE α expression across species. Preferential translation initiation at ATG⁴⁶ could be another causative factor contributing to high level expression of RhoE α . We added the new analyses and discussed the issue in the revised manuscript.

Kozak sequence	G	C	C	A	C	C	A	T	G	G
ATG ¹	A	C	A	T	C	A	A	T	G	A
ATG ⁴⁶	T	C	T	A	T	C	A	T	G	G

Figure 3E. Comparison of Kozak sequences around human RhoE translation initiation sites. The strongest Kozak context promoting translation initiation contains a purine, preferably adenine A, 3 bp before ATG, and a guanine G following ATG.

Human	C	A	T	C	A	A	T	G	A	A	G	G	A	G	A	G	A	G	A	G	C	C	A	G	C	C	A	G	A	A	T	T	A	T	C	C	A	G	C	A	A	T	C	T	A	T	C	A	T	G	G	A	T	C	C	T	A	
Gorilla	C	A	T	C	A	A	T	G	A	A	G	G	A	G	A	G	A	G	A	G	C	C	A	G	C	C	A	G	A	A	T	T	A	T	C	C	A	G	C	A	A	T	C	T	A	T	C	A	T	G	G	A	T	C	C	T	A	
Cow	G	A	T	C	A	A	T	G	A	A	G	G	A	G	A	G	A	G	A	G	C	C	A	G	C	C	A	G	A	A	T	T	A	T	C	C	A	G	T	A	A	A	T	C	T	A	T	C	A	T	G	G	A	T	C	C	T	A
Mouse	C	A	T	C	A	A	T	G	A	A	G	G	A	G	A	G	A	G	A	G	C	C	A	G	C	C	A	G	A	A	T	T	A	T	C	C	A	G	T	A	A	A	T	C	T	A	T	C	A	T	G	G	A	T	C	C	T	A
Xenopus	C	A	T	C	A	A	T	G	A	A	G	G	A	A	G	A	A	G	A	A	C	G	A	G	T	C	A	G	A	A	C	T	G	G	C	C	A	G	C	A	A	T	C	A	A	T	G	A	T	G	G	A	T	C	C	C	A	
Zebrafish	A	A	T	C	A	A	T	G	A	A	G	G	A	G	A	G	A	G	A	T	C	T	A	T	C	C	A	G	A	A	C	T	G	T	C	C	C	T	T	C	A	A	C	C	C	G	T	A	A	T	G	G	A	T	C	C	G	A

Figure 3F. Conservation analysis for RhoE α translation start site among species.

a) this leads to preferential translation initiation at ATG46 and needs to be addressed in the manuscript. In addition to “RhoE α being more stable with a longer half-life than RhoE”, as the authors discuss, the preferential translation would be a major factor explaining the observed higher expression level of RhoE α ;

Response: This is an insightful suggestion. We are truly thankful. The sequence analyses (Fig. 3E and 3F) along with discussions were added in the revised manuscript.

b) the sequence comparison shown in Fig 2 should include an additional 5 nucleotides 5' to ATG1 so that this Kozak effect can be recognized by the reader;

Response: Excellent point. We revised the figure accordingly (Figure 3F).

Human	C A T C A A T G A A G G A G A G A A G A G C C A G C C A G A A A T T A T C C A G C A A A T C T A T C A T G G A T C C T A
Gorilla	C A T C A A T G A A G G A G A G A A G A G C C A G C C A G A A A T T A T C C A G C A A A T C T A T C A T G G A T C C T A
Cow	G A T C A A T G A A G G A G A G A A G A G C C A G C C A G A A A T T A T C C A G T A A A T C T A T C A T G G A T C C T A
Mouse	C A T C A A T G A A G G A G A G A A G A G C C A G C C A G A A A T T A T C C A G T A A A T C T A T C A T G G A T C C T A
Xenopus	C A T C A A T G A A G G A A A G A A G A A C G A G T C A G A A A C T G G C C A G C A A A T C A A T G A T G G A T C C C A
Zebrafish	A A T C A A T G A A G G A G A G A A G A T C T A T C C A G A A A C T G T C C C T T C A A C C C G T A A T G G A T C C G A

Figure 3F. Conservation analysis for RhoE α translation start site among species.

c) Fig 3 C and D seem to indicate that in the absence of ATG46, the full-length RhoE form (that starts at ATG1) is drastically reduced. Is this a matter of stability of RhoE protein? Can this be rescued by co-transfecting construct 3 and 4 from Fig 3C (i.e. RhoE and RhoE α) into the RhoE-KO Hela cells?

Response:

This is an interesting phenomenon. To rule out if there is any direct effect of RhoE α on the expression of RhoE or RhoE protein stability, we re-introduced RhoE α in the cells and assessed the level of RhoE protein by Western blot. We found that forced expression of RhoE α had a minimal impact on RhoE protein level (Fig. 4D).

Figure 4D. Forced expression of RhoE α had minimal impact on RhoE protein level.

Since two initiation sites ATG¹ and ATG⁴⁶ is not far away, loss of the integrity of ATG⁴⁶ Kozak motif (e.g. ATG⁴⁶>AAA) may weaken the translation initiation complex formation at ATG¹ and consequently reduce RhoE expression as shown in Fig. 3C and 3D.

3) Concerning the functional divergence between RhoE and RhoE α

a) This part of the manuscript lacks any experimental validation of the differentially expressed genes, which were identified by RNAseq. Quantitative PCR needs to be done for at least some candidate genes, otherwise these experiments are not very meaningful. In particular

highlighting pathways that are very distant from the cell types used, including autoimmune response or liver cell metabolism, make the value of this analysis questionable. Bioinformatic filtering of sequence data can yield very different outcomes and the mere listing of functional categories as presented in this part of the manuscript is neither informative nor helpful for the isoform characterization.

b) concerning the relative contribution of this RNAseq data interpretation to the remaining manuscript, this part is of excessive length. In addition, the lack of informativity of Figs 5 and 6 would suggest to present them as supplemental material and include a new Figure 5 that presents some experimental validation of suspected candidate genes.

Response to a and b: We totally agree with the reviewer's comments and understand the limitation of the data. By taking reviewer's suggestion, we made three major revisions: 1) organ- and/or tissue-specific signaling was filtered out in bioinformatic analysis; 2) RNA-seq data were validated by qPCR; 3) limitation of the study was discussed. Finally, Fig. 5 and 6 were moved to supplemental section.

c) page 9, first sentence (also page 3), the authors state that they compared the molecular signaling profiles of the two proteins; however, they analyzed their gene expression profiles, which is not the same. This should be corrected.

Response: We appreciate the comments and corrected accordingly.

d) It might be interesting to add a simple anti-flag co-immunoprecipitation experiment comparing both isoforms individually with respect to the described interaction with effectors such as Syx and p190RhoGAP.

Response: Yes, that is a good suggestion. We added co-IP experiments and demonstrated that RhoE α remained binding capability to Syx, p190RhoGAP and ROCK1. The new data were presented as Fig. 5A and 5C.

Figure 5A. ROCK1 binds to both RhoE and RhoE α .

Figure 5C. Both RhoE and RhoE α binds to Syx and p190RhoGAP.

minor corrections:

5) page 3 last paragraph: “and this site is located at the downstream of the classical RhoE translation start site” should be “and this site is located downstream of the classical RhoE translation start site.”

Response: Yes, we corrected as suggested.

6) page 5, paragraphs on Confocal imaging and on Cycloheximide (CHX) chase assay: the sentence “HeLa cells were seeded in a 6-well plate for overnight and subsequently (..)” is not accurate and should be corrected, e.g. to “HeLa cells were seeded in a 6-well plate, incubated overnight and subsequently (..)”

Response: Yes, we corrected as suggested.

7) Figure 1F: although the MS-identified peptides unequivocally belong to Rho E, the two drawn lines pointing to the excised bands do not seem to indicate the bands at 25 and 27 kDa; please check carefully, especially since the antibodies do not seem to be very specific or efficient in RhoE immunoprecipitation

Response: We thank the reviewer’s careful attitude and agreed that majority of published data including ours showed RhoE protein band higher than 25kDa. The error was corrected.

8) page 7, second paragraph: in the sentence “Alternative translation initiation and alternative splicing from a single gene are the two major mechanisms resulting in a generation of protein isoform in most circumstances^{28, 29}”, the cited references do not include work from the alternative splicing field. Second, it would be advisable to refer at this stage that another possibility for the observed isoform band pattern could be a post-translational modification such as phosphorylation that frequently causes a shift in electrophoretic mobility.

Response: Yes, we added one original⁷ (Ref. 30 in manuscript) and one review article⁸ (Ref. 31 in manuscript) regarding alternative splicing as suggested. The PTM issue was also discussed and referenced.

9) page 7 and Figure 2: The authors argue that the transfected RhoE cDNA also yields the expression of two bands in cells, thus indicating alternative translation initiation. It is important to state more clearly how this cDNA construct was cloned and what sequences from its 5´ or 3´ untranslated regions were included. The corresponding sentence in the materials and method section is currently: “RhoE cDNA (ENST00000263895.8) was amplified by PCR from HeLa cDNA and inserted into pcDNA3.1 or COOH-3xflag vector.” but Fig 2 indicates the presence of 5-utr sequence.

Response: We thank the reviewer for pointing out this description inaccuracy and revised the text accordingly. Basically, 5´ UTR was included in the expression vectors expressing both RhoE and RhoE α . 5´ UTR was not included in vectors expressing RhoE or RhoE α individually.

10) page 7: grammar error: “and the alternative translation start site should be located at the downstream of ATG1. should be: “and the alternative translation start site should be located downstream of ATG1”.

Response: Yes, we have corrected as suggested.

11) page 8: style: “Collectively, these data provide evidence that RhoE α is a new isoform of RhoE with 15 amino acids shorter compared to RhoE.” this would be better as : “...that the new isoform RhoE α is 15 amino acids shorter compared to RhoE.”

Response: Yes, we have revised it accordingly.

1. Fiegen D, Haeusler LC, Blumenstein L, Herbrand U, Dvorsky R, Vetter IR and Ahmadian MR. Alternative splicing of Rac1 generates Rac1b, a self-activating GTPase. *J Biol Chem*. 2004;279:4743-9.
2. Yap K, Xiao Y, Friedman BA, Je HS and Makeyev EV. Polarizing the Neuron through Sustained Co-expression of Alternatively Spliced Isoforms. *Cell Rep*. 2016;15:1316-28.
3. Riento K, Totty N, Villalonga P, Garg R, Guasch R and Ridley AJ. RhoE function is regulated by ROCK I-mediated phosphorylation. *EMBO J*. 2005;24:1170-80.
4. Riou P, Kjaer S, Garg R, Purkiss A, George R, Cain RJ, Bineva G, Reymond N, McColl B, Thompson AJ, O'Reilly N, McDonald NQ, Parker PJ and Ridley AJ. 14-3-3 proteins interact with a hybrid prenyl-phosphorylation motif to inhibit G proteins. *Cell*. 2013;153:640-53.
5. Garg R, Koo CY, Infante E, Giacomini C, Ridley AJ and Morris JDH. Rnd3 interacts with TAO kinases and contributes to mitotic cell rounding and spindle positioning. *J Cell Sci*. 2020.
6. Fiegen D, Blumenstein L, Stege P, Vetter IR and Ahmadian MR. Crystal structure of Rnd3/RhoE: functional implications. *FEBS Lett*. 2002;525:100-4.
7. Modrek B, Resch A, Grasso C and Lee C. Genome-wide detection of alternative splicing in expressed sequences of human genes. *Nucleic Acids Res*. 2001;29:2850-9.
8. Modrek B and Lee C. A genomic view of alternative splicing. *Nat Genet*. 2002;30:13-9.

REVIEWERS' COMMENTS:

Reviewer #1 (Remarks to the Author):

The authors have provided a revised manuscript and have done a good job to answer the reviewers' questions: added a number of controls and new experiments, toned down some conclusions and considered additional explanations. However, two points initially raised by myself remains.

- Considering my previous point #2, the issue remains. The authors should not talk about "a new isoform" before figure 3A.

Page 7 lane 6, the sentence "The mass sequences of the two protein bands highly matched to RhoE protein, confirming the two isoforms of RhoE » should be toned down as « The mass sequences of the two protein bands highly matched to RhoE protein, compatible with the existence of two isoforms of RhoE »

- For new Figure 5B, showing the impact of expressing RhoE and RhoEa on the actin cytoskeleton, the anti-flag staining corresponding to the transfected cells should be shown on the figure as an additional channel.

Minor comments:

- page 11 lane 15-17, the authors cite a publication about the regulation of Rnd3 expression by miRNAs. This citation is very restrictive as many other studies (at least 10) demonstrated such a regulation.

- The way the mCherry-RhoEa construct was made should be described in the Materials and Methods section.

Reviewer #2 (Remarks to the Author):

The authors were largely unresponsive to the initial review requesting functional, mechanistic and/or translational insight. Therefore, novelty is limited here, especially since another RhoGTPase (Rac1b) has already been shown to have an alternatively spliced isoform. Enthusiasm is diminished because the manuscript remains preliminary and largely descriptive.

Reviewer #3 (Remarks to the Author):

In general, the major concerns have been adequately addressed by the authors, including the suggested additional experiments.

I suggest a few minor changes:

1) the newly introduced references 25 and 26, which refer to already reported alternative splicing variants of Rho GTPases, would more appropriately cite the original discoveries, namely: Jordan P, Brazão R, Boavida MG, Gespach C and Chastre E (1999). Cloning of a novel human Rac1b splice variant with increased expression in colorectal tumors. *Oncogene* 18, 6835-6839. Marks P. W., Kwiatkowski D. J. Genomic organization and chromosomal location of murine Cdc42, *Genomics*, 1996, 38: 13-18.

2) The requested validation of the differential gene expression levels observed in cells expressing either RhoE or RhoEa has now been included for 4 genes. The two genes shown in Supplementary Fig 3 are referred to in the manuscript text, and a corresponding sentence should also be included

for the other two genes tested and shown in Suppl Figure 4.

In addition, the authors should make the complete list of genes from the RNAseq data available as a supplementary table.

3) Sentences that require correction of typing errors or phrasing:

page 8, line 22: translation, not traslation

page 13:

a) line 15: The data indicate that one-third of genes are commonly regulated by the two proteins and the rest of majorities are differentially regulated by RhoE and RhoEa individually.

either say: ‘..and the remaining are differentially regulated’ or say ‘while the majority is differentially regulated’

b) line 25. investigated, not invested

We thank the reviewers and editor for offering those helpful suggestions on improving the quality of the manuscript. The followings are the responsive summarizations pertaining to the reviewer's specific comments point-by-point.

Reviewer #1:

The authors have provided a revised manuscript and have done a good job to answer the reviewers' questions: added a number of controls and new experiments, toned down some conclusions and considered additional explanations. However, two points initially raised by myself remains.

- Considering my previous point #2, the issue remains. The authors should not talk about "a new isoform" before figure 3A.

Response: Yes, we have replaced "new isoform" with immunoblot or protein as suggested before figure 3a.

Page 7 lane 6, the sentence "The mass sequences of the two protein bands highly matched to RhoE protein, confirming the two isoforms of RhoE » should be toned down as « The mass sequences of the two protein bands highly matched to RhoE protein, compatible with the existence of two isoforms of RhoE »

Response: Yes, the change is made accordingly.

- For new Figure 5B, showing the impact of expressing RhoE and RhoE α on the actin cytoskeleton, the anti-flag staining corresponding to the transfected cells should be shown on the figure as an additional channel.

Response: Yes, an anti-flag imaging (red) is added into Figure 5b.

Minor comments:

- page 11 lane 15-17, the authors cite a publication about the regulation of Rnd3 expression by miRNAs. This citation is very restrictive as many other studies (at least 10) demonstrated such a regulation.

Response: More references are cited as suggested.

- The way the mCherry-RhoE α construct was made should be described in the Materials and Methods section.

Response: Yes, more details of cloning of mCherry-RhoE α construct is added in the Methods.

Reviewer #3:

In general, the major concerns have been adequately addressed by the authors, including the suggested additional experiments.

I suggest a few minor changes:

1) the newly introduced references 25 and 26, which refer to already reported alternative splicing variants of Rho GTPases, would more appropriately cite the original discoveries, namely:

Jordan P, Brazão R, Boavida MG, Gespach C and Chastre E (1999). Cloning of a novel human Rac1b splice variant with increased expression in colorectal tumors. *Oncogene* 18, 6835-6839.

Marks P. W., Kwiatkowski D. J. Genomic organization and chromosomal location of murine Cdc42, *Genomics*, 1996, 38: 13-18.

Response: The two references are added as reference 25 and 26, respectively.

2) The requested validation of the differential gene expression levels observed in cells expressing either RhoE or RhoE α has now been included for 4 genes. The two genes shown in Supplementary Fig 3 are referred to in the manuscript text, and a corresponding sentence should also be included for the other two genes tested and shown in Suppl Figure 4.

In addition, the authors should make the complete list of genes from the RNAseq data available as a supplementary table.

Response: The correlated text is added as suggested. The entire RNA-seq results are deposited in the Gene Expression Omnibus. The accession number is GSE132718.

3) Sentences that require correction of typing errors or phrasing:

page 8, line 22: translation, not trasnlation

page 13:

a) line 15: The data indicate that one-third of genes are commonly regulated by the two proteins and the rest of majorities are differentially regulated by RhoE and RhoE α individually.

either say: ‘..and the remaining are differentially regulated’ or say ‘while the majority is differentially regulated’

b) line 25. investigated, not invested

Response: The appropriate changes are made as suggested. We are thankful for the reviewer’s carefulness, which significantly improves the quality of the manuscript.